The nectar report: quantitative review of nectar sugar concentrations offered by bee visited flowers in agricultural and non-agricultural landscapes

Pamminger Tobias tobias.pamminger@basf.com
Becker Roland
Himmelreich Sophie
Schneider Christof W.
Bergtold Matthias
Global Ecotoxicology, BASF SE , Limburgerhof , Germany
Colla Sheila
Electronic publication date: 2019 Feb 27
Publication date: 2019
Volume: 7
Electronic Location ID: e6329
Received 2018 Oct 15; Accepted 2018 Dec 21
Copyright: ©2019 Pamminger et al.
Copyright year: 2019
Copyright holder: Pamminger et al.
License: This is an open access article distributed under the terms of the Creative Commons Attribution License, which permits unrestricted use, distribution, reproduction and adaptation in any medium and for any purpose provided that it is properly attributed. For attribution, the original author(s), title, publication source (PeerJ) and either DOI or URL of the article must be cited.
License URL: https://creativecommons.org/licenses/by/4.0/

Keywords: Nectar quality, Hymenoptera, Flower resources

Funding: The authors received no funding for this work.

==============================
There is growing concern that some bee populations are in decline, potentially threatening pollination security in agricultural and non-agricultural landscapes. Among the numerous causes associated with this trend, nutritional stress resulting from a mismatch between bee nutritional needs and plant community provisioning has been suggested as one potential driver. To ease nutritional stress on bee populations in agricultural habitats, agri-environmental protection schemes aim to provide alternative nutritional resources for bee populations during times of need. However, such efforts have focused mainly on quantity (providing flowering plants) and timing (during flower-scarce periods), while largely ignoring the quality of the offered flower resources. In a first step to start addressing this information gap, we have used literature data to compile a comprehensive geographically explicit dataset on nectar quality (i.e., total sugar concentration), offered to bees both within fields (crop and weed species) as well as outside fields (wild species) around the globe. Social bees are particularly sensitive to nectar sugar concentrations, which directly impact calorie influx into the colony and consequently their fitness making it an important resource quality marker. We find that the total nectar sugar concentrations in general do not differ between the three plant communities studied. In contrast we find increased variability in nectar quality in the wild plant community compared to crop and weed community, which is likely explained by the increased phylogenetic diversity in this category of plants. In a second step we explore the influence of local habitat on nectar quality and its variability utilizing a detailed sunflower (Helianthus annuus L.) data set and find that geography has a small, but significant influence on these parameters. In a third step we identify crop groups (genera), which provide sub-optimal nectar resources for bees and suggest high quality alternatives as potential nectar supplements. In the long term this data set could serve as a starting point to systematically collect more quality characteristics of plant provided resources to bees, which ultimately can be utilized by scientist, regulators, NGOs and farmers to improve the flower resources offered to bees. We hope that ultimately this data will help to ease nutritional stress for bee populations and foster a data informed discussion about pollinator conservation in modern agricultural landscapes.

Introduction

Pollinators are an integral part of natural as well as agricultural ecosystems, with the majority of flowering plants relying on their ecosystem services (Ollerton, Winfree & Tarrant, 2011). Over the past decades bee pollinators have received particular attention, following the realization that some populations seem to be declining (Biesmeijer et al., 2006; Potts et al., 2010; Ollerton et al., 2014). While managed honey bee populations are declining in only a few geographic regions and over certain time periods (Moritz & Erler, 2016), the focus of concern has recently extended to wild bees (Roulston & Goodell, 2011; Goulson et al., 2015; Vaudo et al., 2015). Numerous potential drivers for this proposed dynamic have been put forward including changes in land use, agricultural intensification, habitat loss or fragmentation and emerging pathogens (Brown & Paxton, 2009; Winfree et al., 2009; Goulson et al., 2015). While all these factors likely contribute to some degree, changes in flower provided food resource for bees has emerged as prime candidate directly regulating bee populations (Roulston & Goodell, 2011). Bees and their larvae almost exclusively rely on flower derived nutrients, namely nectar as their primary source of carbohydrates and pollen for protein, lipids, and other micronutrients essential for development, health and survival (Michener, 2000; Brodschneider & Crailsheim, 2010; Roulston & Goodell, 2011). Large scale changes in land-use can alter the quality, abundance and availability of relevant flower derived resources, which in turn can result in nutritional mismatch leading to nutritional stress for bee populations with potential adverse effects (Potts et al., 2010; Roulston & Goodell, 2011; Goulson et al., 2015). For example, while bee pollinated crops might provide a plethora of flower derived resources during their flowering period, the lack of alternative food sources in monocultural dominated agricultural settings, might put a strain on bee species foraging outside the flowering period.

In order to ease nutritional stress on manages as well as wild bee populations in agricultural settings the establishment of complementary foraging habitats has been incentivized via agro-environmental management schemes in the EU and elsewhere (Phillips & Lowe, 2005; Vaughan & Skinner, 2008; Lye et al., 2009; Goulson et al., 2015; Potts et al., 2015). Such schemes were originally intended to provide bees with complementary flower resources outside the mass flowering periods of commercial crops, but have traditionally been intended to support social bees i.e., Bombus sp. (Vaudo et al., 2015). Only recently the effects of nutritional enhancement on wild bees which often have different habitats and nutritional requirements have come more into focus (Scheper et al., 2015). Besides quantity and timing the quality of floral resources, including total sugar content and sugar concentration have direct fitness consequences for social bees (Brodschneider & Crailsheim, 2010; Vaudo et al., 2015; Vaudo et al., 2016), but likely also solitary bees at least to some degree. Consequently qualitative aspects of nectar resources should be taken into consideration when developing management plans for complementary and high quality nutritional bee resources (Vaudo et al., 2015).

As a first step to facilitate the integration of flower resource quality in pollinator management we have used literature data to compile a geographically explicit database of nectar quality (measured as total sugar concentration) provided by bee visited flowers in an agricultural and natural setting. Given that nectar is the main carbohydrate source for adults as well as developing bees, sugar concentration is directly linked to the amount of sugar bees can extract from flowers and has traditionally served as a proxy for nectar quality (Roulston & Goodell, 2011; Vaudo et al., 2015).

We use the compiled database to compare the quality and quality variability of nectar resource bees can encounter in agricultural landscapes in- (crop and weeds) and off-field (wild) around the globe. In a second step we utilize a unique historical data set to analyze the influence of local habitat and water stress on nectar sugar concentrations and their variability. In a last step we identify crop genera, which provide sub-optimal nectar quality and suggest plant groups which could be used to nutritionally support bee populations in agricultural landscapes during times of need.

Materials and Methods

Data collection and categorization

In late 2017 and early 2018 we searched the literature for records on nectar quality in bee pollinated flowers using ISI web of knowledge and google scholar. We used the search terms: flower AND nectar AND sugar concentration adding either pollinator or bee as additional search term. Using these results, we identified relevant publications by scanning the title and abstract. Based on this refined list we extended our search to the literature cited within the relevant publications. Following the first scan of the primary English literature online, we search for older German primary as well as secondary literature (Books) in our company internal library.

Plant selection

Plant species were categorized as bee visited if either bee pollination was directly observed or the flowers were explicitly classified as “melittophil” based on their floral characteristics by the study authors. In addition, we used the USDA pollinator manual (McGregor, 1976) and the expertise of BASF plant experts for cross validation of the derived classifications.

Geographic localization

We chose to map the plant distribution on a continental scale because this information was available for the majority of plant species included in the data set. We decided to choose the Panama Canal as separation line between North and South America the Ural and the black sea to separate Europe from Asia and the Suez Canal to separate Asia and Africa. Using the encyclopedia of life (http://eol.org/) as source for plant distribution we recorded the presence and absence of collection records of each plant species on the five continents. This very broad geographical classification is intended as a first attempt to make this information geographically explicit and should serve as a starting point to add more detailed information on the local geographic (e.g., national or region) or habitat characteristics in the future. Such information will be vital to make more precise predictions about the temporal quality dynamics in agricultural landscapes around the globe.

Categorization of crop, weed and wild plants

The selected plants were categorized as crop species if they were listed as “cultivated crops” in any of the available governmental databases (e.g., USDA: https://plants.usda.gov and European commission plant variety catalogue: https://ec.europa.eu, McGregor, 1976), the open primary literature or were known as such to our BASF crop experts. All remaining plants without such record were categorized as non-cultivated. In a second step these non-cultivated plants were classified either as a weed species, in case they were listed in at least one of the following agricultural or governmental weed data resource (USA Noxious weed data base https://plants.usda.gov, Australia weeds http://www.environment.gov.au or industry compendium (Bayer, 1992)), or as wild plants in case they were not mentioned in one of these data bases. Once a plant species was categorized (as crop weed or wild) in one geographic region it was classified as such in all other regions where it was present.

Resource quality

We used sugar (total carbohydrate) concentration in nectar (%w/w) as proxy for nectar quality. This quality characteristic was chosen because it is the most frequently reported quantitative measurement of nectar quality in the literature, and is directly related to bee fitness (Vaudo et al., 2015). However, it is important to mention that other quality criteria (e.g., sugar composition, nectar volume as well as the presence and absence of non-sugar compounds) are also important markers for resource quality (Vaudo et al., 2015). In particular nectar volume is likely a secondary main driver for nectar quality, combined with the sugar concentrations it determines the total caloric value per flower. However, such information is scarce and was consequently not included in this project.

Nectar quality categorization

Nectar serves as the main carbohydrate source for bees and consequently the total caloric value as well as the rate of calorie uptake are important aspects of nectar quality for them. One of the main factors determining uptake rate is nectar viscosity, which in term is largely determined by nectar sugar concentration. Based on uptake measurements and theoretical consideration the bee optimal concentration range was determined as 35–65% (Kim, Gilet & Bush, 2011). While this is a theoretical optimal range and bees seem to prefer higher over lower nectar sugar concentrations (Wykes, 1952; Roubik & Buchmann, 1984; Cnaani, Thomson & Papaj, 2006) they will collect nectar with sugar concentrations below that value under natural conditions (e.g., Roubik & Buchmann, 1984). However, available evidence suggests that at least social bees avoid foraging on nectar sources below 20% sugar concentration, likely because the caloric intake cannot support sustained foraging activity with potentially detrimental effects for the bee colony (Maurizio & Grafl, 1980; Roubik & Buchmann, 1984; Cnaani, Thomson & Papaj, 2006). While most of these results are based on findings in social bees (honeybee and bumblebee) we can assume that most of the basic physiological limitations (energy expenditure during flight and physics of suction feeding) apply to solitary bees as well. Based on these criteria we define nectar concentrations of 65–35% as optimal 35–20% as adequate and nectar sugar concentrations below 20% as low quality.

Analysis

Nectar quality and its variability in bee visited plants

In the first part of the analysis we focused on the broad picture of nectar quality and its variation provided by a given plant community (crop, weeds and wild) on all relevant continents around the globe. In addition, we explore the possibility of intrinsic differences in nectar quality variability of the plant species belonging to the different communities (crop, weed and wild) using plant species where we had multiple quality measurements (N > 3) to calculate standard deviation (SD) as a proxy for within species variability.

The influence of local habitat on nectar quality and its variability

During our non-english literature screening we discovered a data set (Simidtschiev, 1988), which is uniquely suited to isolate the contribution of geographic location and water availability to nectar sugar concentrations and its variation in the sunflower (Helianthus annuus L.). In order to make this data more easily accessible to the scientific community, we will give a brief summary of the materials and methods used. Between 1981 and 1986 a field experiment was conducted at two field sites in Bulgaria (Toshevo in north-east and Plovdiv in central Bulgaria) separated by more than 300 km. Over this time period 52 sunflower variants and hybrids, originating from different geographical regions around the globe (including Europe, North America, South America and Australia), were grown under standard agronomical conditions at both locations. The nectar sugar concentration for all varieties was measured each year on 25 flowers per variant/hybrid day and location (200–300 measurements per year) using a capillary based extraction method and an Abbe Refractometer. In a second experiment, the author tested the effect of irrigation (watering vs. no watering) using a subset of four varieties. Using this unique data set (Simidtschiev, 1988) we explore the influence of location on nectar sugar concentration variation in the sunflower taking advantage of the paired design of the study.

Nectar quality offered by plant genera

In a last step we compared the quality of crop genera in terms of nectar quality. We used all genera, where we had measurements for more than 3 plant species belonging to this genus. We characterized the selected genera according to our pre-defined categories (see above) as optimal (35–65%), adequate (34–20%) and low quality (below 20%). We used this information to identify crop genera offering low quality nectar and potential genera offering high nectar quality as potential replacements.

Statistics

Both statistical analysis and graphs generation were conducted in R v. 3.3.3. (R Core Team, 2013). We used descriptive statistics, conservative non-parametric Kruskal–Wallis (KW) and the Fligner-test (FT) to explore the overall differences between the three plant communities both in terms of nectar sugar concentration (KW) and its variation (FT) on a global level and within the geographic regions. In case the main test indicated significant differences, a Bonferroni corrected pairwise test (KW or FT) was used. To explore differences in the nectar quality variability on a species level between communities (crop, weed and wild) we used a KW test. In order to test for the influence of the geographic location of cultivation (Simidtschiev, 1988 data set) on nectar concentrations and its variation we used a paired Wilcoxon-test as well as a FT test. To investigate the last hypothesis of genus specific differences in nectar quality a KW test was used. This approach was chosen to present the overall patterns in nectar quality and its variation, which does not take into account the phylogenetic dependencies of the individual plant species. In case this data would be used to identify potential drivers for the observed variation a phylogenetically controlled approach would be more appropriate. However, the main focus of this paper is the presentation of the overall broad patterns, while an in depth analysis of the factors driving it were beyond the scope of this project. Significance level were set to α = 0.05 in all cases.

Results

Data summary

In total we collected 444 individual measurements of sugar concentration in nectar for bee pollinated flowers ranging from 6.3–85%. With similar sampling sizes for plant species in crop (N = 151) and wild plants (N = 141), but fewer measurements for weeds (N = 30). On a genus level we find that the wild community has the highest phylogenetic diversity in terms of number of genera recorded (N = 63) followed by the crop community (N = 29) and lowest diversity in the weed community (N = 18). In general, the recorded data is evenly spread across the geographic regions (see Table 1), however only a limited number of weed species could be identified in Africa (N = 13) and South America (N = 18). The summary statistics including mean, median 10th and 25th percentile are presented in Table 1.

Table 1 Summary statistic of the sugar concentration (%) of crop, weed and wild plant communities across the globe.

Region	Community	N	Median	Mean	10th Percentile	25th Percentile	
Global	ALL	322	40	41.0	25	32	
	Crop	151	39.7	39.2	24	32	
	Weed	30	39.8	41.6	30.5	33.6	
	Wild	141	41	32	25	32	
Europe	ALL	236	39.7	40.3	24.9	32.3	
	Crop	144	39.9	39.4	24.1	32.9	
	Weed	30	39.8	41.6	30.5	33.6	
	Wild	62	39.3	41.7	24.9	29.9	
North America	ALL	240	40	40.9	25	32.7	
	Crop	145	40	39.6	24.1	33	
	Weed	30	39.8	41.6	30.5	33.6	
	Wild	65	44	43.8	25.3	32.3	
South America	ALL	234	40	41.3	26.4	33.9	
	Crop	136	40	40.8	28.3	34.9	
	Weed	18	41.7	42.5	32.9	37	
	Wild	80	40	42	26	32	
Africa	ALL	168	41	41.9	27.5	34.9	
	Crop	133	40	40.9	28.5	35	
	Weed	13	40	42.8	32.8	34	
	Wild	22	51.8	47.6	23.3	34	
Asia	ALL	211	40	40.9	26	33	
	Crop	141	40	40	25.8	34	
	Weed	25	43.4	42.5	31.7	34.7	
	Wild	45	40	42.9	25.4	32.3	
Australia	ALL	203	40	40.9	26	33	
	Crop	139	40	40	25.8	43	
	Weed	24	39.8	41.6	31.4	33.4	
	Wild	40	40.8	43.6	25.9	32.5	

Nectar quality & variability

Overall nectar concentration in all regions were comparable around a median value of 40% sugar concentration (see Fig. 1, Table 1) and no significant differences between crop, weed or wild plant communities were found globally (KW chi2 = 3.2, p = 0.2) or within the different geographic regions (all KW chi2 <4.48, p > 0.11; see Fig. 1 and Table 1). In contrast to the median concentrations we find that the three plant communities differed in the variability of nectar quality (Global community; Fligner test chi2 = 31.97, p < 0.001). This effect is mainly driven by an increased variability of the wild community (see Fig. 1) which differs significantly from the crop community on a global level (Bonferroni corrected pairwise Fligner test crop × wild chi2 = 30.64 , p < 0.001), with a similar trend in the same direction when compared to the weed community (Bonferroni corrected pairwise Fligner test; crop × weed chi2 = 5.02 , p = 0.08). In contrast we find that crop and weed species clearly do not differ in terms of their variability (Bonferroni corrected pairwise Fligner test weed × wild chi2 = 1.01, p = 0.93). When comparing the variability of nectar quality on a species level we find that we had only a limited number of species with multiple nectar measurements (N > 2) recorded (crop N = 18, weed N = 6 and wild N = 18). Using this limited data set we find no indication of intrinsic difference in variability (measured as SD) of plant species belonging to the three different plant communities (KW chi2 = 2.52, p = 0.28).

Figure 1 Summarizes the total nectar sugar concentration (%) in agricultural landscapes on a global as a continental level.

We present data for Europe, North America, South America, Africa, Australia and overall (Global) for crop (A–G), weed and wild plant communities. Results of the statistical analysis (Kruskal Wallis (KW) chi2, and p values) are presented in the upper left corner of the individual panels.

Regional effects on nectar quality and its variability

When reanalyzing the Simidtschiev (1988) sunflower dataset comparing nectar sugar concentrations we find that geographic location has a small (MedianToschevo = 30.7%, MedianPlovdiv = 36.05% see Fig. 2), but significant influence on nectar concentrations (paired Wilcoxon test: V = 216, p < 0.0001 see Fig. 2) and its variation (Fligner test chi2 = 6.12, p = 0.01 see Fig. 2). When looking at the effect of non-irrigation (natural rainfall) on nectar sugar concentration of the four tested sunflower varieties the original analysis of Simidtschiev (1988) concludes that in three of the four varieties watering did not significantly influence nectar sugar concentration and in the fourth cultivar (hybrid 260) it only decreased it by about 3.4% (mean irrigation = 50.3, mean natural rainfall = 53.7%).

Figure 2 Nectar sugar concentrations of 52 sunflower varieties grown at two locations in Bulgaria.

The graph depicts the nectar sugar concentration of 52 sunflower varieties (Helianthus annuus) grown at two geographic locations in Bulgaria (Toschevow and Plovdiv area) between 1981 and 1986. Data taken and re analyzed from Simidtschiev (1988). Results of the statistical analysis (paired Wilcoxon test, V and p values) are presented in the lower left corner.

Nectar quality on a genus level

In total we recorded multiple measurements for 12 crop and 16 non-crop genera and find that there is a significant difference in nectar sugar concentration between them (KW chi2 = 149.23, p < 0.0001 Fig. 3). When comparing the nectar quality according to our categorization (see above) our results indicate that two crop genera, namely Capsicum (including paprika and chili) and Pyrus (pear), offer low quality nectar (median sugar concentration <20% (see Fig. 3). When looking at the genera offering high quality nectar we were able to identify 15 Genera which provide optimal nectar concentrations (35%–65%) for bees (see Fig. 3).

Figure 3 Nectar sugar concentration on a genus level.

The graph shows the distribution of nectar sugar concentration among all plant genera for which more than three measurements were available. The red lines indicate the boarder of the optimal (65–35%), adequate (35–20%) and low sugar nectar concentration (<20%). Crop genera are marked in red. Results of the statistical analysis (Kruskal Wallis (KW) chi2, and p values) are presented in the lower left Corner.

Discussion

In this study we have compiled the first comprehensive data set on nectar quality provided by bee visited plants in agricultural landscapes around the globe. Our data indicates that nectar sugar concentrations in bee visited flowers is strongly conserved across all communities and geographic regions with a median value around 40% (see Fig. 1 and Table 1). In addition, we find that wild plants exhibit stronger variation in nectar concentrations at the community level when compared to crop plants. However, this difference is not reflected on the species level. Using a comprehensive data set from the German literature (Simidtschiev, 1988) on sunflower varieties we find evidence that microhabitat (e.g., water availability) and geographic region might have a more limited effect on nectar sugar concentration and its variation than previously thought. Using the complete data set we identify two crop genera (Capsicum and Pyrus) which provide low quality nectar to bees during the flowering period and suggest 15 possible genera which provide high quality nectar as potential supplement nectar source.

When looking at the recorded nectar sugar concentrations we find strong support for the well-established idea that flowers are under strong selection pressure to provide nectar suitable for their respective pollinators (Baker, 1975; Harder, 1986; Perret et al., 2001). In the case of bees, the literature suggests optimal values ranging from 35–65% which is well supported by our data (Waller, 1972; Harder, 1986; Kim, Gilet & Bush, 2011). It has been suggested that the pollinator preferences for different sugar concentrations could be explained by different modes of nectar intake, which in case of bees favours higher viscosity and consequently sugar concentrations (Kim, Gilet & Bush, 2011). Our results therefore indicate that nectar quality between different plant communities in agricultural landscapes and their surroundings are (1) closely matched to pollinators needs (2) comparable between all regions and (3) in principal likely sufficient to maintain healthy bee pollinator populations.

In contrast to the median sugar concentrations we find elevated nectar quality variability of the wild plant community compared to the crop and likely weed plants communities (see Fig. 1). However, based on the results of our limited species level data set we have no indication that this pattern is reflected on the species level where plants of the different communities exhibited comparable variability. In particular the species belonging to the weed community are interesting in this regard as they are wild species which grow under (invade) standardized agricultural conditions. A priori we could expect that the more standardized growing conditions in agricultural could reduce nectar quality variability when compared to natural habitats. As we do not find differences between these three groups on the species level our results support the view that growing conditions might have a more limited influence on the variability of nectar sugar concentration and supports a plant species specific nectar concentration of at least some species. In turn this suggest that the observed variation on the community level likely reflects the elevated phylogenetic diversity in wild plants compared to crop and weed species (Meyer, DuVal & Jensen, 2012). Indeed, when looking at the community level, we find the wild community containing more than twice the number of wild genera (N = 63) compared to the crop (N = 29), as well as the weed community (N = 18), which seems the most likely explanation for the observed pattern.

The comprehensive study of Simidtschiev (1988) offers the unique opportunity to study the effects of geography and water availability on nectar sugar concentration. These results support our initial findings that geography, in this case only has a limited absolute effect on nectar sugar concentration and its variation (See Fig. 2). In particular the fact that plants “defended” their nectar sugar concentrations against variation in water availability suggest that at least sunflowers have plant species specific nectar sugar concentrations. In his original analysis Simidtschiev (1988) suggests that instead of changes in nectar concentration, nectar volume responds to reduced water availability, which in turn reduce the caloric value and consequently resource quality for bees. It would be interesting to analyses what parameters best explain the observed variation (including temperature, rainfall soil types ect.) because such factor undoubtedly play a role in shaping nectar concentrations to some degree (Corbet et al., 1979). Unfortunately, these parameters were not recorded by Simidtschiev (1988) for the study duration. This highlights the importance of multiple measurements to adequately characterize resource quality for bees. In a next step it would be very important to include such measurements into the data base to provide a more detailed picture of nectar quality to serve as a robust basis to improve the resource quality offered to bee populations in the future.

In our quality analysis we have identified two genera of crop plants, which provide low quality nectar sugar concentrations. While it is well-known that pears (Pyrus) are not considered attractive to social bees due to their low nectar (Fig. 3) quality (Maurizio & Grafl, 1980) the even lower levels in Capsicum (e.g., paprika and chilies) suggest that these crops will likely not be able to sustain managed bee populations on their own. Such information, coupled with detailed information regarding geographic abundance of these crops, could be used to identify potential targets for a resource quality intervention such as agro-environmental scheme. We suggest 15 plant genera which provide adequate nectar quality for bees and could be used as feeding supplement. For example, pear orchards might be a good option for such interventions as the supplemental nectar resources could be located close by, which in turn could attract and retain social as well as solitary bees which use pear flowers as a source of pollen (Westrich, 1989).

Our study is a first step towards the integration of resource quality in bee conservation practices. The obvious next step could be to include additional quality markers for nectar including nectar volume, sugar composition and non-sugar components, which clearly play an important role in determining nectar quality for bees (Vaudo et al., 2015). In particular nectar volume and flowering season are promising next steps to get a more complete picture of nectar quality because together concentration and volume define the caloric value of individual flowers, entire plants and ultimate vegetation types. The caloric value of flowers is likely important for most bee species, however most studies directly linking nectar quality to bee fitness stem from social bees such as honey and bumblebees. Currently it is less clear how important these factors are for solitary bees’ fitness considering their deviating ecology. Such information would be useful to specifically address the needs of solitary bees. A second important step would be to compile a similar data set for the second important flower resource pollen and its quality markers such as crude protein content, amino acid composition or lipid content (Roulston, Cane & Buchmann, 2000). Such information should be combined with detailed geographic and information on exact flowering periods in order to estimate the resource availability and quality during the season in a given location and plan and implement targeted interventions to support bee populations.

Such a tool could support farmers, scientists, regulators NGOs and the industry when designing optimized alternative flower resources for bees in agricultural landscapes. We hope that this data set will serve as a starting point to help facilitate a data informed discussion about pollinator conservation in agricultural landscapes between all relevant stakeholders and will ultimately help to reduce nutritional stress for bee populations in modern agricultural landscapes.

Conclusion

In this study we conducted the first systematic review of nectar sugar quality bees can encounter in agricultural landscapes in- and off-field around the globe. We report that nectar sugar concentrations do not vary between regions or habitats with a median of around 40% nectar total sugar concentration. We have identified several crop genera providing nectar with sub optimal sugar concentrations for bees which could potentially benefit from alternative nectar sources during their mass flowering period. This dataset is only a first step toward integrating nectar sugar concentrations into bee management practices and we hope that this data resource will facilitate communication between all relevant stakeholders and ultimately help to reduce nutritional stress for bee populations in modern agricultural landscapes.

Additional Information and Declarations

Competing Interests

Author Contributions

Data Availability

All authors work for BASF SE (an agricultural solution provider) and declare that they have no competing interests.

Tobias Pamminger conceived and designed the experiments, performed the experiments, analyzed the data, contributed reagents/materials/analysis tools, prepared figures and/or tables, authored or reviewed drafts of the paper, approved the final draft.

Roland Becker contributed reagents/materials/analysis tools, authored or reviewed drafts of the paper, approved the final draft, contributed ideas.

Sophie Himmelreich, Christof W. Schneider and Matthias Bergtold authored or reviewed drafts of the paper, approved the final draft, contributed ideas.

The following information was supplied regarding data availability:

Pamminger, Tobias (2018): 2018_Pamminger_nectar_sugar_concentrations_peerJ _R1_corr. figshare. Dataset. https://doi.org/10.6084/m9.figshare.7376420.v1.

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
