# Peer review of "The nectar report: quantitative review of nectar sugar concentrations offered by bee visited flowers in agricultural and non-agricultural landscapes"

_PeerJ, doi:10.7717/peerj.6329_

## Round 0.1 · original submission · Major Revisions

Please address the comments by the two reviewers. Be sure to be precise in terms of what pollinator species, taxa or groups you are referring to throughout the manuscript. I look forward to receiving the revised version.

Reviewer 1 ·

Basic reporting

Pamminger et al present an extensive dataset on nectar concentrations in 322 species of wild, cultivated and weedy plant species from across the major world continents. The paper is well structured and well written throughout, with only a few areas that need to have their phrasing polished up which I detail below.

The results are interesting (a very consistent average nectar concentration found in insect pollinated plants across all regions) and highlight particular crop genera that may benefit from additional nectar sources in the surrounding landscape to benefit bee populations.

The figures and tables are clear and the statistical techniques used are well described. The literature is reasonably well covered, but I would like more discussion of the evidence base as to why the sugar concentration in nectar is important, and which species of bee this evidence is drawn from (predominantly social bees). I go into this point in more detail in section 3.

Experimental design

The research question is well defined (the importance of sugar concentrations in nectar) but as detailed in the next section I would like some more discussion of whether or not this is valid to apply to all bees.

Validity of the findings

I have a few more major comments that I would like to see addressed. The major point is that the manuscript talks about the importance of nectar to 'bees', but almost all the evidence we have that nectar concentrations are important comes from social bees (Bombus, Apis, etc). I think the data presented is really interesting, but I find it hard to argue that nectar concentrations are quite so important for solitary bee species (the vast majority of bee species) that have mainly had their fitness linked to the type and quantity of pollen available to them. I think a slight rewrite that shifts the focus (and justification for choosing nectar as the response variable) onto social bees would make the manuscript more representative of our current understanding of how important nectar is to bees, given the substantial knowledge deficit that exists for solitary bee species.

L69-71. I would argue strongly that maintaining a diverse bee community has not been the focus of agri-environment schemes targeted at bees. Early work in the UK focused strongly on bumble bees and their foraging requirements Carvell et al 2004; 2006; 2007 leading to currently incarnations of flower-rich field margins containing predominantly bumble bee food plants (e.g. Trifolium, Centaurea, Vicia and so on). Until recently, few pollinator-focused schemes were appraised for their ability to provide resources to a wide variety of wild bee species e.g. Scheper et al 2015 JAE

L72-75. I think you need to be careful here. Most of the work that has be conducted on floral resource quality (especially nectar) has been on social bees. Next to nothing is known about how nectar quality affects solitary bee fitness, and if it even does at all. Pollen quality has been more extensively studied, but still predominantly on social bees (Vaudo, recent work on polyfloral/monofloral pollen diets for B. terrestris colonies from the Michez lab). For solitary bees, the limited amount of work suggests that it is the quality of the landscape (i.e. density of floral resources) that determines bee fitness e.g Persson et al 2018 Ecology and Evolution. I would lay out a couple of these issues in a little more detail in the introduction to highlight that most of the nectar work comes from social bees, which are of course very important as pollinators due to their abundance in most ecosystems.

L152. This dataset from Kim et al 2011 is restricted to a narrow selection of bees in Apinae, e.g. Bombus, Apis, Meliponini and Euglossini, all but the Euglossines being social bees. These bees, whilst capturing a lot of the social diversity, are not necessarily representative of the other 19,000+ bee species and so statements such as “However, all evidence suggests that bees avoid foraging on nectar sources below 20% sugar concentration, likely because the caloric intake cannot support sustained foraging activity with potentially detrimental effects for the bee colony” are not necessarily true for solitary species. An earlier discussion of the ecological differences between social and solitary bees seems warranted. If the paper was to state that the focus on nectar was to better inform management for social bees (as is implied by the focus on nectar and references to colony health) this would resolve many of these issues. As it is currently written, the paper refers to ‘bees’ but focuses mostly on social bee examples and on a problem (nectar concentration) that has not empirically been demonstrated to be an issue in solitary species.

L326-327. I was interested to read this reference to see what evidence the authors present, but it is incomplete and gives only the title of the book without publisher information. Looking it up online, the subtitle is ‘the most important food sources for the honey bee’. As written, the current sentence “While it is well-known that pears (Pyrus) are not considered attractive to bees due to their low nectar (Fig.3) quality (Maurizio and Grafl 1980)” is misleading, since the evidence is for honey bees only, and pear is known to have attractive pollen for spring-flying wild bees, particularly Andrena (Chambers 1946 J Animal Ecology; Westrich 1989 Die Wildbienen Baden-Württembergs). Focusing the manuscript more explicitly on honey bees, bees known to be sensitive to nectar concentrations, would seem more appropriate.

Additional comments

Formatting of references – I have not seen the first and second author et al citation structure before. Please check against PeerJ standards

L11. “some bee populations are in decline,” add comma
L14. “associated with this trend, nutritional stress resulting from a mismatch between bee nutritional needs and plant community provisioning has been suggested as one potential driver.” change placement of commas
L19. “this information gap,” add comma
L42. Are in decline, or seem to be declining
L44. rephrase to ‘are declining in only a few geographic regions and over certain time periods’
L48. correct to “intensification, habitat loss- or fragmentation and emerging pathogens” check use of commas in lists
L60. cornucopia – perhaps a more scientific and slightly less flowery word? (no pun intended)
L61. Discreet – perhaps change to relatively short?
L65. I’m not quite sure what you mean by ‘selective’ in this context. Is it the habitat that is selective, or the criteria used to determine the kind of habitat type that is selective? ‘Specific’ may be a more appropriate word choice if you intend the latter meaning
L82-83. Add comma ‘developing bees, sugar concentration’
L86. Check second quality
L126-133. Check formatting e.g. double parentheses, underlining etc
L147-149. Repetition of ‘for bees’
L213. Check capitalisation of genus
L223. Check PeerJ formatting for Tab. or Table 1
L248. Check spelling of variety
L249. Check spelling of hybrid
L254. Difference singular
L273. Check spelling of Pyrus
L308. Repetition of findings, consider rephrasing
L331-335. Repetition of high-quality, consider rewriting. In fact, both sentences could be rewritten for clarity.
L347. Check capitalisations

Figure 2. Check spelling of Wilcoxon
Figure 3 legend. Check spelling of border
Figure 3. Hoffmannseggia and Hoffmanseggia seems to be listed twice, are these different genera?

Reviewer 2 ·

Basic reporting

The manuscript is generally well-written and well structured in professional English and easy to read. The introduction is convincing and presents the background information necessary to understand the objectives of the paper. Additionally, the figures are relevant and well described.
However, the raw data has not been supplied. It is necessary to include a list with all species for which nectar sugar concentrations were found and the references. This list could be in the appendix.

Experimental design

The study is well designed and the approach is appropriate for the research questions. However, some important information is missing, for example the exact search terms for the search in ISI web of knowledge and google scholar. Additionally, all data bases that were searched should be mentioned to make it possible to repeat the study.
In the method section on the statistical analysis some important information is missing, for example which test was used to test which hypothesis. Moreover, I think that it is difficult to compare the variability of nectar sugar concentration between crops, weeds and wild plants. The authors write themselves that the higher variability in wild plants is probably due to the fact that this group includes more taxa. Unless the authors do not account for the number of taxa included in the analysis I don’t think that it makes sense to test for these differences.

Validity of the findings

The authors have compiled a large data set of reported nectar sugar concentrations from the published literature. Therefore, the findings are robust and the conclusions are supported by the data. However, I wonder how important the sugar content really is for bees. For example a mass flowering crop might provide such a high nectar availability that the sugar content might be completely unimportant. These constraints could be more emphasized in the discussion. Additionally, I miss some more concrete ideas how the data is useful for pollinator conservation. For example, I like the idea very much to combine this data set on nectar quality with the timing of flowering in plants. Why was that not done in this study? It could be very useful to know when most crop/wild plant species are not flowering to support plants that provide nectar of high quality in times of scarcity.

Additional comments

L21 insert "from the literature"
L80 compiled from the literature
L81-82 Did you really only study plants from agricultural landscapes? I understood that you integrated all published reports of nectar quality including those from wild plants. Are some of these plants also growing for example in forests and not in agricultural landscapes?
L85 replace "this data base" by "the data base that we compiled"
L97-98 You should state the exact search codes to make the study replicable.
L99-101 How was that done? Which literature was searched?
L126-127 You should state all data bases that were used.
L131-133 Which data base was used for Europe? You should give the details on all data bases used.
L141-144 Is there any data how important each of those variables is for nectar quality?
L152 Can you explain why the highest sugar concentrations (above 65%) are not optimal for bees? Another questions that I have is why plants would produce nectar concentrations that are above the optimum for bees. Do they attract other species?
L158 You write about bee colonies, so are these results only about honey bees?
L176 replace "possible" by "possibility"
L182 I do not understand what was "paired". Irrigated vs. non-irrigated fields?
L199-206 More details are needed. Which test was used for which hypothesis using which data set?
L203 heterogeneity?
L215 replace "the lest" by "and we found the lowest"
L235 replace by "we had only a limited number of species with multiple..."
L238 the three different groups?
L248 In line 179 you write that there were 52 different varieties?
L264 See comment on agricultural landscapes above.
L273 Pyrus
L294-296 This hypothesis should be stated in the introduction.
L309 "in ab"?
L325-329 Do you know of any evidence that this low nectar quality is balanced by high quality pollen resources?
L333-335 I think this is important for every pollinator dependent crop, not only orchards.
L341-343 You should give some references.
L356 See comment on agricultural landscapes above.

---

## Round 0.2 · Minor Revisions

Please incorporate the few remaining recommendations by the two reviewers. Many thanks.

Reviewer 1 ·

Basic reporting

The manuscript is well written, and the revision has improved the format and structure of the references. There are a couple of areas where the English could be polished that I have notes in the general comments to the authors.

Following a comment from Reviewer 2, the data are now accessible via a download link.

The overall thrust of the article is clear: to describe global patterns of nectar quality and to use this information to propose suitable plant species for management interventions to support farmland bee populations.

Experimental design

Unchanged from the first review, meets all PeerJ requirements for scope and clarity of research question.

Validity of the findings

The authors have now added more discussion to all parts of the manuscript on the subject of sugar concentrations in nectar and how this is important for solitary bee species, as opposed to just social bee species. It is a difficult area to discuss since most of the empirical work has been conducted on social bee species, but I am happy that the manuscript now sets this knowledge gap in a wider context for readers. Overall, I am happy for this manuscript to be accepted with only the very minor changes I detail below.

Additional comments

I just have a couple of minor places where the English could be tightened up a little for readability.

L17. (during flower-scarce periods)
L21. suggest rephrase as “as well as outside fields (wild species)”
L21-22. suggest rephrase as “Social bees are particularly sensitive to”
L23-24. suggest rephrase as “We find that the total sugar concentrations in nectar generally do not differ between the three plant communities studied”
L24-25. In contrast,
L25. Increased variability in nectar quality
L32. Database is one word; do you mean data set?


L353. resources could be located close by, which in turn
L354. change to “which use pear flowers as a source of pollen”

Reviewer 2 ·

Basic reporting

The basic reporting is good.

Experimental design

The experimental design is sound and now described in more detail. I would just like to know whether the second experiment of Simidtschiev et al. (1988) with the irrigation treatment was also conducted in two regions or only in one. In case it was also performed in both regions, region should be included in the model either as a fixed or random effect.

Validity of the findings

The findings are valid.

Additional comments

The manuscript has improved compared to last version. Many parts are now much more clear and I only have few minor comments:

L79-80 replace by "taken into consideration when developing management plans for complementary and high quality nutritional bee resources."
L138 replace by "in case they were not mentioned in one of these data bases"
L171 replace by "below 20% as low quality"
L195-196 Was this second experiment also conducted in two regions? In this case region should be included in the analysis either as a random or fixed effect.
L203 "pre-defined categories"
L209 add citation of R software
L210 delete "used" in this line, otherwise it is confusing which test was used for which question
L215 make clear what you mean by "communities"
L216 make clear what you mean by "local habitat"
L229 You should mention at this point where your data can be found.
L260 I don't think that "local habitat" is the right term here, because I would assume that the plants were grown in different habitat types. Maybe better "Regional effects on nectar quality..."?
L267 replace by "varieties"

---

## Round 0.3 · accepted · Accept

Congratulations. Thank you for clearly addressing the suggested revisions.

#